# PARP Inhibitors in Glioma: A Review of Therapeutic Opportunities

**DOI:** 10.3390/cancers14041003

**Published:** 2022-02-16

**Authors:** Hao-Wen Sim, Evanthia Galanis, Mustafa Khasraw

**Affiliations:** 1NHMRC Clinical Trials Centre, University of Sydney, Sydney, NSW 2050, Australia; hao-wen.sim@svha.org.au; 2St Vincent’s Clinical School, University of New South Wales, Sydney, NSW 2010, Australia; 3Department of Medical Oncology, The Kinghorn Cancer Centre, Sydney, NSW 2010, Australia; 4Department of Medical Oncology, Chris O’Brien Lifehouse, Sydney, NSW 2050, Australia; 5Department of Oncology, Mayo Clinic, Rochester, MN 55905, USA; galanis.evanthia@mayo.edu; 6Duke University School of Medicine, Duke University, Durham, NC 27710, USA

**Keywords:** poly(ADP-Ribose) polymerase inhibitors, DNA damage, brain cancer, glioma, glioblastoma

## Abstract

**Simple Summary:**

There is a pressing need for new effective treatments against glioma. A promising option is a class of drugs called poly(ADP-Ribose) polymerase (PARP) inhibitors. PARP inhibitors block the repair of DNA damage. They may work synergistically with radiotherapy, chemotherapy and immunotherapy. They may work particularly well in settings where other DNA damage repair pathways are impaired. This article reviews the clinical trials investigating PARP inhibitors in glioma.

**Abstract:**

Gliomas are the most common malignant primary brain tumor in adults. Despite advances in multimodality therapy, incorporating surgery, radiotherapy, systemic therapy, tumor treating fields and supportive care, patient outcomes remain poor, especially in glioblastoma where median survival has remained static at around 15 months, for decades. Low-grade gliomas typically harbor *isocitrate dehydrogenase* (*IDH*) mutations, grow more slowly and confer a better prognosis than glioblastoma. However, nearly all gliomas eventually recur and progress in a way similar to glioblastoma. One of the novel therapies being developed in this area are poly(ADP-Ribose) polymerase (PARP) inhibitors. PARP inhibitors belong to a class of drugs that target DNA damage repair pathways. This leads to synthetic lethality of cancer cells with coexisting homologous recombination deficiency. PARP inhibitors may also potentiate the cytotoxic effects of radiotherapy and chemotherapy, and prime the tumor microenvironment for immunotherapy. In this review, we examine the rationale and clinical evidence for PARP inhibitors in glioma and suggest therapeutic opportunities.

## 1. Introduction

The majority of malignant primary brain tumors in neuro-oncology practice are adult-type diffuse gliomas. They arise from neuroglial cells which surround and support the neurons within the central nervous system. As per the 5th edition of the World Health Organisation Classification of Tumors of the Central Nervous System (CNS WHO), adult-type diffuse gliomas are classified by combined histological and molecular features into: astrocytomas which are *isocitrate dehydrogenase* (*IDH*)-mutant, CNS WHO grades 2–4; oligodendrogliomas which are *IDH*-mutant and 1p/19q-codeleted, CNS WHO grades 3–4; and glioblastomas which are *IDH*-wildtype, CNS WHO grade 4 [1].

Unfortunately, all adult-type diffuse gliomas are incurable. There is a paucity of therapeutic options; for instance, the FDA has not approved any new systemic agents for glioblastoma since temozolomide in 2005 [2]. Median survival has remained static around 15 months for decades. Beyond clinical trials, the standard of care for glioblastoma continues to be maximal safe neurosurgery, followed by radiotherapy with concomitant and adjuvant temozolomide. Additionally, there is variable access to alternating electric field therapy. There is limited evidence that salvage therapies prolong survival. Options include repeat neurosurgery, radiotherapy, temozolomide, lomustine and bevacizumab [3]. Gliomas affect the full spectrum of ages including children and young adults. They are associated with high morbidity and complex care needs, including physical disability, cognitive impairment and caregiver stress. Evidently, there is a pressing need for novel therapies against gliomas.

Poly(ADP-Ribose) polymerase (PARP) inhibitors showed activity against *BRCA*-associated malignancies. Subsequent efforts have been made to extend this to the neuro-oncology setting. In this review, we introduce the class of PARP inhibitors and current clinical applications; the importance of brain-penetrant drugs in the neuro-oncology setting; the rationale to combine PARP inhibitors with radiotherapy, chemotherapy and immunotherapy; the rationale to use PARP inhibitors within selected subgroups such as *IDH*-mutant gliomas; and review current and prospective clinical trials. From this, we offer a perspective on future opportunities to advance neuro-oncology practice using PARP inhibitors.

## 2. Targeting DNA Damage Repair

The importance of DNA damage repair has been alluded to in gliomas. For instance, the efficacy of temozolomide, a mainstay chemotherapy option for gliomas, is dependent on the functional status of DNA damage repair systems [4]:Temozolomide produces O^6^-methylguannine adducts which are incorrectly paired with thymine instead of cytosine during DNA synthesis. O^6^-methylguanine-DNA-methyltransferase (MGMT) is a DNA repair enzyme which removes the abnormal methyl groups to neutralize this DNA damage. MGMT promoter methylation reduces the expression of MGMT, which in turn improves the efficacy of temozolomide. Conversely, unmethylated tumors are associated with temozolomide resistance.Mismatch repair systems facilitate the abnormal pairing of O^6^-methylguannine adducts with thymine. They need to be intact for temozolomide-induced cytotoxicity. Conversely, mismatch repair deficiency leads to temozolomide resistance.

Building on this premise, the class of PARP inhibitors is the most established of the DNA damage response modifiers [5]. They are understood to interfere with DNA damage repair via several mechanisms: inhibition of base excision repair; trapping of PARP-DNA complexes; and impaired processing of Okazaki fragments, which are small fragments of newly synthesized DNA (Figure 1). PARPs also have in role in homologous recombination, non-homologous end joining and alternative end joining [6]. In turn, this leads to genomic instability, catastrophic DNA damage and cell death.

Inhibition of base excision repair: Single-strand DNA damage is repaired by mismatch repair, nucleotide excision repair or base excision repair, whereas double-strand DNA damage is repaired by homologous recombination or an error-prone system of non-homologous end-joining [7]. The class of PARP inhibitors canonically interferes with the base excision repair pathway [8]. Ordinarily, PARP enzymes bind to sites of DNA damage, catalyze the attachment of polymers of ADP-Ribose in a process called PARylation, then recruit and regulate effector repair proteins. The catalytic activity of the PARP enzymes are blocked by PARP inhibitors. The base excision repair pathway is essential to repair damaged bases caused by ionizing radiation and alkylating agents such as temozolomide.Trapping of PARP-DNA complexes: PARP inhibitors also lead to the trapping of PARP1 at sites of DNA damage. These persistent PARP-DNA complexes interfere with DNA replication. They promote collapse of the replication fork, which is the region where the DNA helix is unwound and used as a template for DNA synthesis, in turn leading to double-strand DNA breaks [8]. Alternatively, the replication fork may reverse direction when encountering PARP-DNA complexes, then accelerate in an unrestrained manner to cause double-strand DNA breaks [9]. New generation PARP inhibitors such as talazoparib efficiently trap PARP-DNA complexes and have potent cytotoxicity [10]. This appears to be independent of blocking the catalytic activity of the PARP enzymes.Impaired Okazaki fragment processing: Finally, PARP inhibitors may interfere with the processing of Okazaki fragments [11]. During replication, the leading DNA strand (forward direction) is replicated in a continuous manner. However, the lagging DNA strand (reverse direction) is replicated in an interrupted manner as Okazaki fragments, since DNA polymerase can only work in a forward direction. These Okazaki fragments are later connected together by DNA ligase. Ordinarily, PARP1 acts as a key sensor for these unbound Okazaki fragments. Consequently, disruption by PARP inhibitors causes gaps to form within the lagging DNA strand. This cytotoxicity appears to be independent of the previous mechanisms.

When exposed to PARP inhibitors, cancer cells will accumulate single-strand DNA breaks and collapsed replication forks. Despite this, the resulting double-strand DNA breaks can ordinarily be salvaged by the homologous recombination pathway. Synthetic lethality, whereby the effects of PARP inhibitors are combined with homologous recombination deficiency (HRD), may be necessary for cell death [8]. For example, *BRCA1/2* is integral to the homologous recombination pathway. For *BRCA*-associated malignancies, patients harbor a germline *BRCA1/2* mutation in one allele throughout all cells, then lose the second allele exclusively within cancer cells as an obligate step of carcinogenesis. HRD due to loss of both *BRCA1/2* alleles within cancer cells, and thus synthetic lethality when combined with PARP inhibitor use, will be tumor-specific and spare normal cells. This was evidenced by a landmark phase 1 trial of olaparib monotherapy in patients with *BRCA*-associated malignancies, which demonstrated potent tumor-specific activity in a clinical setting [12]. Currently, the PARP inhibitors which have established clinical application are olaparib, rucaparib, niraparib and talazoparib (Table 1). Olaparib is used in ovarian cancer, *BRCA*-mutant breast cancer, *BRCA*-mutant pancreatic cancer and *HRD*-deficient prostate cancer [13,14,15,16,17,18,19,20,21,22,23]. Rucaparib is used in ovarian cancer and *BRCA*-mutant prostate cancer [24,25,26,27,28], and there are promising data for *BRCA*-mutant pancreatic cancer [29]. Niraparib is used in ovarian cancer [30,31,32,33] and talazoparib is used in *BRCA*-mutant breast cancer [34].

There are a variety of other genes with important functions in the homologous recombination pathway, including *ARID1A*, *ATM*, *ATRX*, *BAP1*, *BARD1*, *BLM*, *BRIP1*, *CHEK1/2*, *FANCA/C/D2/E/F/G/L*, *MRE11A*, *NBN*, *PALB2*, *RAD50*, *RAD51*, *RAD51B* and *WRN* [35]. For gliomas, it is notable that *IDH* mutations may induce a phenotype similar to *BRCA1/2* mutations [36]. Cancer cells with these defects and corresponding HRD may be susceptible to PARP inhibitor use. Additionally, cancer cells with a high level of replication pressure will undergo greater DNA turnover and damage. This may overwhelm the homologous recombination pathway, heralding sensitivity to PARP inhibitor use [37]. Finally, combinations of radiotherapy, chemotherapy and immunotherapy with PARP inhibitors can lead to synergistic activity [37].

Evidently, the development of PARP inhibitors to date has been focused on monotherapy use in tumors harboring *BRCA1/2* mutations or other HRD, which are uncommon in gliomas. Instead for gliomas, as expanded below, there may be opportunities to expand the treatment repertoire using PARP inhibitors in combination with radiotherapy, chemotherapy and immunotherapy, and in selected subgroups such as *IDH*-mutant glioma (Figure 2).

## 3. Blood Brain Barrier Uptake

A major impediment to drug development in gliomas has been the blood brain barrier. This is a physiological barrier consisting of tight junctions between cerebral endothelial cells, and a dynamic system of transporters including P-glycoprotein (P-gp) and breast cancer resistance protein (BCRP) which act as efflux pumps [38]. They limit the accumulation of neurotoxic substances within the brain parenchyma. However, they can also restrict entry of pharmacologic drugs such as PARP inhibitors. An important prerequisite for therapeutic action is that the drug makes it to the target tissues; for gliomas, this entails circumventing the blood brain barrier.

A historical narrative has been that drug exposure in gliomas should be adequate, as contrast enhancement on imaging yields evidence of blood brain barrier disruption. However, this disruption is regional and heterogeneous, and infiltrating glioma cells extend well beyond the margins of contrast enhancement [39]. It has been noted that the concentration of many drugs is significantly lower at the infiltrating rim compared to the necrotic tumor core. The use of dedicated brain-penetrant drugs for gliomas can be a critical determinant of efficacy.

Regarding clinically approved PARP inhibitors, the pharmacokinetics of olaparib in glioblastoma has been investigated in the OPARATIC trial [40]. Suitable patients with recurrent glioblastoma received olaparib prior to neurosurgical resection. Olaparib concentrations were then measured in tumor core and tumor margin by mass spectrometry. Olaparib was detected in 71/71 tumor core specimens (27 patients) and 21/21 tumor margin specimens (9 patients) at radiosensitizing concentrations, thus demonstrating reliable penetration. In a pharmacokinetic study based on mouse tumor xenograft models, niraparib was compared to olaparib [41]. Niraparib was found to have greater tumor exposure, sustainability within the brain, and tumor growth inhibition than olaparib.

Conversely, several studies have suggested that rucaparib and talazoparib may have limited blood brain barrier penetration. In vitro and in vivo experiments found that rucaparib was efficiently eliminated by P-gp and BCRP transporters [42]. A pharmacokinetic study based on patient-derived glioblastoma xenograft models showed that rucaparib had limited accumulation within the tumor or evidence of activity [43]. Similarly, a pharmacokinetic study based on patient-derived glioblastoma xenograft models showed that talazoparib was efficiently eliminated by P-gp overexpression. There was limited accumulation within the tumor and loss of temozolomide sensitization [44].

In terms of other PARP inhibitors, veliparib was found to have a higher brain-to-plasma concentration ratio than either rucaparib or talazoparib, despite liability to P-gp and BCRP transporters [45]. Pamiparib is not a substrate of either P-gp or BCRP. Relative to olaparib, niraparib or talazoparib, pamiparib demonstrated higher penetration across the blood brain barrier in mice [46].

The use of brain-penetrant PARP inhibitors confers greater potential for therapeutic efficacy in gliomas. These pharmacokinetic studies are fundamental to ensure blood brain barrier penetration and should be undertaken early on during the drug development process. In addition, preoperative dosing studies such as the OPARATIC trial help to confirm drug delivery in the clinical setting [40].

## 4. PARP Inhibition with Radiotherapy or Tumor Treating Fields

PARP inhibitors and radiotherapy may combine synergistically via several mechanisms:The base excision repair pathway is essential to repair damaged bases caused by ionizing radiation. The pathway is blocked by PARP inhibitors, which result in sensitizer enhancement ratios of 1.2–1.7 in glioma cells [47]. This radiosensitizing effect appears to increase in a replication-dependent manner, due to greater DNA turnover and damage with higher replication.The persistence of cancer stem cells contributes to radioresistance. One of the mechanisms that allow cancer stem cells to remain viable is the upregulation of DNA damage repair pathways [48]. PARP inhibitors may contribute to abrogating this response [49].Tumor hypoxia contributes to radioresistance. PARP inhibitors may exert a vasodilatory effect, which reduces tumor hypoxia and radioresistance [50]. PARP inhibitors may also have separate anti-angiogenic effects which enhance radiation sensitivity [51,52].Radiotherapy can effectively disrupt the blood brain barrier, augmented by modern localization techniques [53]. This can enhance the delivery of PARP inhibitors into tumor tissues.

The combination of olaparib and radiotherapy was recently investigated in the phase I PARADIGM trial [54]. A total of 16 patients with newly diagnosed glioblastoma aged over 70 were enrolled, and received olaparib with short-course radiotherapy, divided into four dose-escalation cohorts. The recommended phase 2 dose of olaparib with short-course radiotherapy was found to be 200 mg twice daily. Ongoing trials of olaparib and radiotherapy include OLA-TMZ-RTE-01 [55], which is evaluating the recommended phase 2 dose of olaparib with long-course radiotherapy followed by olaparib with temozolomide, and PARADIGM-2 [56], which is evaluating the recommended phase 2 dose of olaparib with long-course radiotherapy, then 4 weeks of olaparib, then 6 months of temozolomide in the *MGMT*-methylated cohort only.

Niraparib is currently being combined with radiotherapy (NCT05076513) or tumor treating fields (TTF) (NCT04221503). Instead of ionizing radiation, TTF uses alternating electrical fields to disrupt the mitotic apparatus which is required for cell division. TTF may also have effects on the homologous recombination pathway and increases blood brain barrier permeability, which complements PARP inhibitor use [57].

The combination of veliparib and radiotherapy has been studied recently. VERTU was a randomized non-comparative phase 2 trial involving 125 adults with newly diagnosed *MGMT*-unmethylated glioblastoma [58]. The experimental arm consisted of veliparib and radiotherapy, followed by adjuvant veliparib and temozolomide. The standard arm consisted of concurrent temozolomide and radiotherapy, followed by adjuvant temozolomide. Although veliparib was well-tolerated, there was insufficient benefit overall. In the experimental arm, the progression-free survival rate at 6 months was 46% and the median overall survival was 12.7 months. Comprehensive biomarker discovery is underway, and we await the outcome of a complementary trial Alliance A071102 (NCT02152982) for patients with newly diagnosed *MGMT*-methylated glioblastoma. PBTC-033 was a phase 1/2 trial which enrolled 65 children with diffuse intrinsic pontine glioma [59]. They received veliparib and radiotherapy, followed by adjuvant veliparib and temozolomide, but their overall survival did not improve relative to historical benchmarks. The 1-year and 2-year overall survival rates were 37% and 5%, respectively. Veliparib and radiotherapy were also combined in a phase 2 trial of newly diagnosed diffuse pediatric-type high-grade glioma, with results pending (NCT03581292).

Pamiparib and radiotherapy were combined in Study 104 [60]. This phase 1/2 trial consisted of multiple dose-escalation and expansion arms. Pamiparib 60 mg twice daily, in conjunction with radiotherapy and temozolomide, was generally well tolerated in patients with newly diagnosed or recurrent glioblastoma. The objective response rate was 8% and the disease control rate was 65%.

Finally, iniparib was added to radiotherapy and temozolomide in a phase 2 trial of 81 patients with newly diagnosed glioblastoma [61]. The activity of iniparib was modest and subsequent investigations indicated that iniparib was not a dedicated PARP inhibitor [62,63].

Table 2 summarizes the ongoing trials involving PARP inhibition with radiotherapy or tumor treating fields. To date, the reported trials have consisted of relatively small heterogenous glioma populations, and there has yet to be convincing evidence of clinical benefit. The accumulating data from ongoing trials will hopefully refine the use of PARP inhibitors in this setting.

## 5. PARP Inhibition with Chemotherapy or Antiangiogenics

Temozolomide remains a mainstay chemotherapy option for gliomas. It is an alkylating agent that reacts with DNA bases to generate a range of potentially cytotoxic lesions, which are usually repaired by the base excision repair pathway. PARP inhibitors block this pathway and may act synergistically with temozolomide. This has been corroborated by several glioma xenograft models, which consistently show a significant delay in tumor growth [64,65,66,67].

Olaparib was combined with temozolomide in the dose expansion phase of the OPARATIC trial [40].

This combination was challenging due to overlapping hematological toxicity. The recommended phase 2 dose of olaparib was de-escalated to 150 mg daily, 3 days per week only. Alternatively, olaparib has been combined with an antiangiogenic drug cediranib in an ongoing phase 2 trial for patients with recurrent glioblastoma (NCT02974621). This avoids overlapping toxicity and there is potential for complementary action. PARP inhibitors may exert antiangiogenic effects on tumor vasculature [51,52]. Antiangiogenic drugs may downregulate *BRCA1/2*, thus disrupting homologous recombination and lead to synthetic lethality with PARP inhibitors [68].

Veliparib has been combined with temozolomide. In addition to the radiotherapy plus veliparib and temozolomide trials mentioned previously, there was a phase 1/2 trial involving 225 patients with recurrent glioblastoma (NCT01026493) [69]. A schedule of temozolomide at 150–200 mg/m^2^ for 5 out of 28 days was far better tolerated than at 75 mg/m^2^ for 21 out of 28 days. The objective response rate was low and the survival outcomes were poor. *MGMT* methylation status was not collected for this study, and there remains interest in Alliance A071102 (NCT02152982) due to evidence of greater veliparib sensitivity in an *MGMT*-methylated population [70]. There was also a phase 1 trial involving 29 children with recurrent primary brain tumors (NCT00994071) [71]. The doses of veliparib and temozolomide were limited by myelosuppression, which was more problematic than anticipated based on adult data.

Table 2 summarizes the ongoing trials involving PARP inhibition with chemotherapy or antiangiogenics. The novel PARP inhibitors fluzoparil (NCT04552977) and NMS-03305293 (NCT04910022) are under investigation. Given the recurring hematological toxicity when combining PARP inhibitors and temozolomide, moving forwards, we may need to prioritize the use of PARP inhibitors in combination with other classes of systemic agents, including antiangiogenics, immunotherapy or targeted agents as appropriate.

## 6. PARP Inhibition with Immunotherapy

There is emerging evidence that PARP inhibitors may enhance immunotherapy:By targeting the DNA damage repair pathways, PARP inhibitors cause an accumulation of DNA damage and genomic instability. This is expected to increase the tumor mutational burden and immunogenicity [72]. The efficacy of immune checkpoint inhibitors in highly mutated tumors has been well established [73].PD-L1 is a co-inhibitory ligand of the immune system which may be a predictive biomarker of response to immune checkpoint inhibitors [74]. PARP inhibitors eventually lead to double-strand DNA breaks within cancer cells, which in turn induce inflammatory pathways to upregulate PD-L1 expression. This includes activation of the cGAS-STING pathway, activation of the ATM-ATR-CHK1 pathway and inhibition of GSK-3β [72,75,76,77].Gliomas are characterized by a highly immunosuppressive environment dominated by glioma-associated macrophages which have limited innate activity [78]. PARP inhibitors contribute to reprogramming of the tumor microenvironment and may promote a favorable Th1-mediated immune response [79]. In particular, PARP inhibitor-mediated activation of the cGAS-STING pathway increases the number of tumor-infiltrating lymphocytes [72]. PARP inhibitors also appear to have other pleiotropic effects on the tumor microenvironment, including production of cytokines which regulate natural killer cells and angiogenesis [72,77,79].

A phase 2 basket trial of olaparib in combination with durvalumab in *IDH*-mutant solid tumors is underway (NCT03991832) [80]. At the time of the interim report, nine patients had been enrolled in the recurrent *IDH*-mutant glioma arm, consisting of two grade 2 gliomas, four grade 3 tumors and three grade 4 tumors. There was a partial response in 1 of 9 patients and the median progression-free survival was 2.5 months.

The pre-clinical rationale here appears to be robust. An increasing variety of novel immunotherapeutics are under development, including bifunctional antibodies targeting TGF-β and other immune regulators, which may augment the synergy with PARP inhibitors. Additional clinical trials in this setting are anticipated.

## 7. PARP Inhibition in *IDH*-Mutant Glioma

*IDH* is a nicotinamide adenine dinucleotide phosphate (NADP+)-dependent enzyme and functions in the tricarboxylic acid cycle to catalyze the oxidative decarboxylation of isocitrate to α-ketoglutarate [81]. *IDH* mutations confer neomorphic activity that converts α-ketoglutarate to the oncometabolite 2-hydroxyglutarate [82]. In turn, this interferes with the homologous recombination pathway and appears to render tumor cells exquisitely sensitive to PARP inhibitors. This is based on in vivo models, whereby this effect can be completely reversed by treatment with IDH inhibitors, and conversely, can be completely re-instituted by treatment with 2-hydroxyglutarate enantiomers in cells with intact IDH proteins [36,83]. The accumulation of 2-hydroxyglutarate also inhibits the AlkB family of DNA repair enzymes which may contribute to synthetic lethality with PARP inhibitors [84,85]. These findings have been confirmed in a comprehensive series of DNA repair functional studies, viability assays and in vivo studies using pamiparib in *IDH*-mutant glioma models [86]. Altogether, this forms the basis for a therapeutic strategy with PARP inhibition within the subgroup of *IDH*-mutant glioma.

Regarding clinical trials to date, olaparib monotherapy has been investigated in the phase 2 OLAGLI trial, which enrolled 35 patients with recurrent *IDH*-mutant high-grade glioma [87]. Olaparib was well-tolerated, with fatigue being the most common adverse event. There was a partial response in 2 of 35 patients and the median progression-free survival was 2.3 months. As mentioned above, there is also an ongoing phase 2 basket trial of olaparib and durvalumab which includes a dedicated recurrent *IDH*-mutant glioma arm (NCT03991832) [80]. Table 2 shows a number of other ongoing trials involving PARP inhibition in *IDH*-mutant glioma, which highlights the significant interest in this therapeutic strategy.

## 8. Future Directions

PARP inhibitors have established themselves as part of the treatment armamentarium against ovarian cancer, *BRCA*-mutant breast cancer, *BRCA*-mutant pancreatic cancer and *HRD*-deficient prostate cancer. In addition, despite significant therapeutic nihilism in neuro-oncology, PARP inhibitors may have a future role in treating gliomas, with a multitude of key clinical trials underway. Novel PARP inhibitors are under development which have demonstrable blood brain barrier penetration. Other DNA damage response modifiers are under development, including inhibitors of ATM (NCT03423628), DNA-PK (NCT02977780) and Wee1 (NCT01849146), which share similar therapeutic principles to the class of PARP inhibitors. In addition, there are efforts to understand the potential mechanisms of resistance to PARP inhibition and ways to circumvent them [88]. For example, there are 17 PARP enzymes with variable function (originally reported as 18 PARP enzymes, but tankyrase 3 is a splice variant of tankyrase 2) [89]. Most inhibitors target PARP1 and/or PARP2, but a range of PARP enzymes may need to be targeted to mitigate compensatory activity. The phase of the cell cycle may alter the effectiveness of PARP inhibition, since homologous recombination occurs in the S and G2 phases. CDK12 inhibition has demonstrated reversal of PARP inhibitor resistance [90]. Other resistance mediators include BET, HSP90, ATM, ATR and c-MET, and attempted blockade of multiple pathways is underway.

To maximize effectiveness, we will need to leverage the synergy between PARP inhibition and radiotherapy, chemotherapy and/or immunotherapy. We will need to focus on key subgroups such as *IDH*-mutant glioma, as it is unlikely that all glioma patients will benefit from a common therapeutic strategy. This is analogous to the experience using PARP inhibition in breast, pancreatic and prostate cancer. Currently, there are limitations with the available clinical trial data, principally due to the heterogeneity in the study populations, variety of treatment regimens, early phase uncontrolled designs and small patient numbers. These make it difficult to draw strong conclusions at this time. To ameliorate these ongoing challenges, comprehensive translational work is underway to identify biomarkers of treatment response and resistance to inform the optimal use of PARP inhibitors, as well as greater adoption of clinical trial infrastructure such as adaptive platform designs and synthetic control arms to expedite the drug development process.

Collectively, PARP inhibitors in glioma represent a promising and much needed area of neuro-oncology research. Regarding the pipeline of glioma trials, there is currently a focus on precision oncology, including upfront molecular profiling of tumors and then targeted therapies, serial molecular profiling of tumors at recurrence to understand tumor evolution and resistance, and analysis of liquid biopsy specimens. There is considerable investment in cellular-based and virus-mediated immunotherapeutics. PARP inhibitors are anticipated to play a complementary role in terms of enhancing the action of these novel targeted and immune therapies by abrogating the common pathway of DNA damage repair. Finally, there have been renewed efforts to interrogate the pharmacokinetics and pharmacodynamics of novel agents in window-of-opportunity and trigger trial designs, and complemented by a range of blood brain barrier disrupting methods such as focused ultrasonography, that may expand the repertoire of brain-penetrant systemic agents including PARP inhibitors.

## Figures and Tables

**Figure 1 cancers-14-01003-f001:**
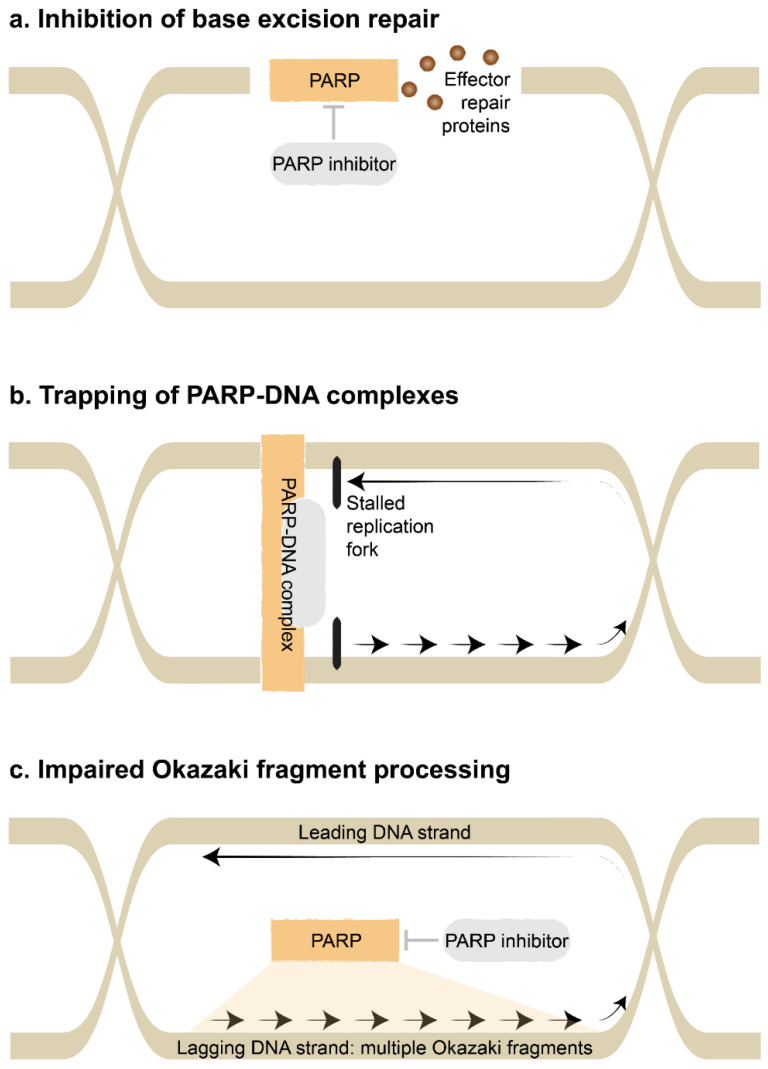
Mechanism of action of PARP inhibitors. (**a**). In the event of cytotoxic DNA damage, PARP binds to the damaged DNA strand and recruits effector repair proteins. This activity is blocked by PARP inhibitors. (**b**). The accumulation of PARP inhibitor-PARP-DNA complexes causes the replication fork to stall and then collapse. (**c**). PARP acts as a chaperone for the multiple Okazaki fragments in the lagging DNA strand during replication. This activity is blocked by PARP inhibitors. Abbreviations: DNA: deoxyribonucleic acid; PARP: poly(ADP-Ribose) polymerase.

**Figure 2 cancers-14-01003-f002:**
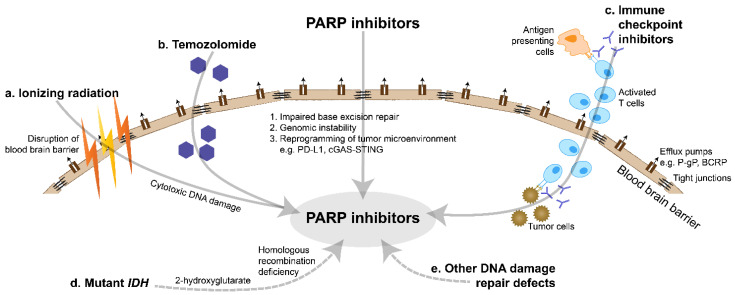
Overview of PARP inhibitor interactions. PARP inhibitors need to circumvent the blood brain barrier, which is characterized by tight junctions and multiple efflux pumps, to achieve therapeutic action. PARP inhibitors combined with radiotherapy, chemotherapy and immunotherapy can lead to synergistic activity. In addition, homologous recombination deficiency and other DNA damage repair defects contribute to synthetic lethality. (**a**). Ionizing radiation causes cytotoxic DNA damage and the base excision repair pathway is blocked by PARP inhibitors. Ionizing radiation also disrupts the blood brain barrier. (**b**). Temozolomide causes cytotoxic DNA damage and the base excision repair pathway is blocked by PARP inhibitors. (**c**). Immune checkpoint inhibitors, including CTLA-4 and PD-1/PD-L1 inhibitors, induce anti-tumor immune responses. Immunogenicity is augmented by PARP inhibitors via genomic instability and reprogramming of the tumor microenvironment. (**d**). *IDH* mutations cause accumulation of the oncometabolite 2-hydroxyglutarate. In turn, this can lead to homologous recombination deficiency and synthetic lethality, in combination with impaired base excision repair due to PARP inhibitors. (**e**). Other DNA damage repair defects, including in the ATM, DNA-PK and Wee1 pathways, can lead to synthetic lethality, in combination with impaired base excision repair due to PARP inhibitors. Abbreviations: BCRP: breast cancer resistance protein; cGAS: cyclic GMP-AMP synthase; *IDH*: *isocitrate dehydrogenase*; PARP: poly(ADP-Ribose) polymerase; PD-L1: programmed death ligand 1; P-gp: P-glycoprotein; STING: stimulator of interferon genes.

**Table 1 cancers-14-01003-t001:** Clinical application of PARP inhibitors.

PARP Inhibitor	Indication	Evidence
Olaparib (AZD2281)	Newly diagnosed advanced ovarian cancer: maintenance treatment after a response to platinum chemotherapy; germline or somatic *BRCA* mutation	SOLO-1 [13,14] olaparib (*n* = 260) vs. placebo (*n* = 131) PFS HR 0.33 (95% CI 0.25–0.43)
	Recurrent ovarian cancer: maintenance treatment (in combination with bevacizumab) after a response to platinum chemotherapy; underlying HRD	PAOLA-1 [15] bevacizumab/olaparib (*n* = 537) vs. bevacizumab/placebo (*n* = 269) overall: PFS HR 0.59 (95% CI 0.49–0.72) HRD subgroup: PFS HR 0.33 (95% CI 0.25–0.45)
	Recurrent ovarian cancer: maintenance treatment after a response to platinum chemotherapy	SOLO-2 [16,17] olaparib (*n* = 196) vs. placebo (*n* = 99) PFS HR 0.30 (95% CI 0.22–0.41) OS HR 0.74 (95% CI 0.54–1.00)
		Study 19 [18,19] olaparib (*n* = 136) vs. placebo (*n* = 129) PFS HR 0.35 (95% CI 0.25–0.49) OS HR 0.73 (95% CI 0.55–0.95)
	Recurrent ovarian cancer: germline *BRCA* mutation; three or more prior chemotherapy regimens	Pooled analysis [20] olaparib (*n* = 205 in primary efficacy population) response rate 31% (95% CI 25–38%)
	Recurrent *HER2*-negative breast cancer: germline *BRCA* mutation	OlympiAD [21] olaparib (*n* = 205) vs. physician’s choice (*n* = 97) PFS HR 0.58 (95% CI 0.43–0.80)
	Newly diagnosed advanced pancreatic cancer: maintenance treatment after a response to platinum chemotherapy; germline *BRCA* mutation	POLO [22] olaparib (*n* = 92) vs. placebo (*n* = 62) PFS HR 0.53 (95% CI 0.35–0.82)
	Recurrent prostate cancer: underlying HRD; prior enzalutamide/abiraterone	PROfound [23] olaparib (*n* = 256) vs. physician’s choice (*n* = 131) overall: PFS HR 0.49 (95% CI 0.38–0.63) *BRCA* subgroup: PFS HR 0.34 (95% CI 0.25–0.47)
Rucaparib (AG014699)	Recurrent ovarian cancer: maintenance treatment after a response to platinum chemotherapy	ARIEL3 [24,25] rucaparib (*n* = 375) vs. placebo (*n* = 189) overall: PFS HR 0.36 (95% CI 0.30–0.45) HRD subgroup: PFS HR 0.32 (95% CI 0.24–0.42) *BRCA* subgroup: PFS HR 0.23 (95% CI 0.16–0.34)
	Recurrent ovarian cancer: germline or somatic *BRCA* mutation; two or more prior chemotherapy regimens	Study 10 [26] rucaparib (*n* = 42 in phase II expansion) response rate 60% (95% CI 43–74%)
		ARIEL2 [27] rucaparib (*n* = 40 in *BRCA*-mutant subgroup) response rate 80% (95% CI 64–91%)
	Recurrent prostate cancer: germline or somatic *BRCA* mutation; prior androgen receptor-directed therapy/taxane chemotherapy	TRITON2 [28] rucaparib (*n* = 62 in primary efficacy population) response rate 44% (95% CI 31–57%)
Niraparib (MK4827)	Newly diagnosed advanced ovarian cancer: maintenance treatment after a response to platinum chemotherapy	PRIMA [30] niraparib (*n* = 487) vs. placebo (*n* = 246) overall: PFS HR 0.62 (95% CI 0.50–0.76) HRD subgroup: PFS HR 0.43 (95% CI 0.31–0.59)
	Recurrent ovarian cancer: maintenance treatment after a response to platinum chemotherapy	NOVA [31,32] niraparib (*n* = 372) vs. placebo (*n* = 181) germline *BRCA* subgroup: PFS HR 0.27 (95% CI 0.17–0.41) non germline *BRCA* subgroup: PFS HR 0.45 (95% CI 0.34–0.61)
	Recurrent ovarian cancer: underlying HRD; three or more prior chemotherapy regimens	QUADRA [33] niraparib (*n* = 47 in primary efficacy population) response rate 28% (95% CI 16–43%)
Talazoparib (BMN673)	Recurrent *HER2*-negative breast cancer: germline *BRCA* mutation	EMBRACA [34] talazoparib (*n* = 287) vs. physician’s choice (*n* = 144) PFS HR 0.54 (95% CI 0.41–0.71)

Abbreviations: 95% CI = 95% confidence interval; HR = hazard ratio; HRD = homologous recombination deficiency; PFS = progression-free survival.

**Table 2 cancers-14-01003-t002:** Ongoing trials of PARP inhibitors in glioma.

Clinical Trial	Phase	Study Population	Intervention	Status
**PARP Inhibition with Radiotherapy or Tumor Treating Fields**				
NCT03212742 [55] (OLA-TMZ-RTE-01)	1/2a	Newly diagnosed glioblastoma	Radiotherapy with olaparib/temozolomide, then olaparib/temozolomide	Recruiting
CRUKD/16/010 [56] (PARADIGM-2)	1	Newly diagnosed glioblastoma	*MGMT*-methylated cohort: Radiotherapy with olaparib/temozolomide, then olaparib/temozolomide *MGMT*-unmethylated cohort: Radiotherapy with olaparib, then olaparib	Recruiting
NCT05076513	0 “trigger”	Newly diagnosed glioblastoma (Cohort A)	Radiotherapy with niraparib, then niraparib	Recruiting
NCT04221503	2	Recurrent glioblastoma	Tumor treating fields with niraparib	Recruiting
NCT03581292	2	Newly diagnosed high-grade glioma; *H3 K27M*-wildtype; *BRAFV600E*-wildtype; children/young adults	Radiotherapy with veliparib, then veliparib/temozolomide	Completed accrual
**PARP Inhibition with Chemotherapy or Antiangiogenics**				
NCT02974621	2	Recurrent glioblastoma	Olaparib/cediranib vs. bevacizumab	Completed accrual
NCT02152982 (Alliance A071102)	2/3	Newly diagnosed glioblastoma; *MGMT*-methylated	Veliparib/temozolomide	Completed accrual
NCT04552977	2	Recurrent glioblastoma	Fluzoparil/temozolomide	Not yet open
NCT04910022	1/2	Recurrent glioblastoma	NMS-03305293/temozolomide	Not yet open
**PARP Inhibition in *IDH*-Mutant Glioma**				
NCT03212274 (ETCTN-10129)	2	Recurrent *IDH*-mutant glioma, cholangiocarcinoma or other solid tumor	Olaparib	Recruiting
NCT05076513	0 “trigger”	Recurrent *IDH*-mutant glioma (Cohort B)	Niraparib	Recruiting
NCT04740190 (TAC-GReD)	2	Recurrent high-grade glioma; DNA damage repair deficiency e.g., *IDH*-mutant, *PTEN*-mutant, BRCAness signature	Radiotherapy with talazoparib/carboplatin	Recruiting
NCT03914742 (ABTC-1801)	1/2	Recurrent *IDH*-mutant glioma	Pamiparib/temozolomide	Recruiting
NCT03749187 (PNOC-017)	1	Recurrent *IDH*-mutant glioma; children/young adults	Pamiparib/temozolomide	Recruiting

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
