# Peer review of "PARP Inhibitors in Glioma: A Review of Therapeutic Opportunities"

_cancers, 2022, doi:10.3390/cancers14041003_

Round 1

Reviewer 1 Report

Glioma PARP trials are still in progress. Wait to write review article of clinical trialsw until effects of PARP treatment on glioma survival can be determined.

Author Response

Reviewer #1

Comment 2.1: Glioma PARP trials are still in progress.  Wait to write review article of clinical trials until effects of PARP treatment on glioma survival can be determined.

Response 2.1: The use of PARP inhibitors is highly topical in the neuro-oncology literature at the present time, with important research being undertaken throughout the translational and clinical trial domains.  Our intent was to present a comprehensive review of the current landscape and therapeutic opportunities of PARP inhibitors in glioma.  Given the known challenges in conducting clinical trials in the neuro-oncology setting, we believe that a comprehensive review on this topic would be highly valuable to the broader clinical and research community   We have renamed our manuscript “PARP inhibitors in glioma: A review of therapeutic opportunities” to emphasize this and to reflect the emerging nature of the literature.

Reviewer 2 Report

The article under review is well written and the ideas have been clearly stated. It also makes an important review of the state of the art and knowledge in the field, having chosen the references wisely. For these reasons I have no major considerations to take into account and I directly recommend the publication of the article.

A minor consideration would be the use of too short and somewhat unconnected sentences in some parts of the paper. For example, between lines 204 and 210. I recommend the use of a less telegraphic style for a more pleasant reading. The tables are also too large and should not occupy more than one page in the final version to achieve an overall view of the information presented.

Beyond these recommendations concerning style, the article provides an accurate overview of the field and suggests new therapeutic approaches that are worthwhile and should be considered and published.

Author Response

Reviewer #2

Comment 3.1: The article under review is well written and the ideas have been clearly stated.  It also makes an important review of the state of the art and knowledge in the field, having chosen the references wisely.  For these reasons I have no major considerations to take into account and I directly recommend the publication of the article.

Response 3.1: Thank you for the favorable feedback.

Comment 3.2: A minor consideration would be the use of too short and somewhat unconnected sentences in some parts of the paper.  For example, between lines 204 and 210.  I recommend the use of a less telegraphic style for a more pleasant reading.

Response 3.2: The sections “Targeting DNA damage repair”, “PARP inhibition with radiotherapy or tumor treating fields” and “PARP inhibition with immunotherapy” were intended to include bulleted lists, but they were incorrectly formatted in the pdf proof.  We have corrected this oversight.

Comment 3.3: The tables are also too large and should not occupy more than one page in the final version to achieve an overall view of the information presented.

Response 3.3: The tables summarize key information as a definitive resource for investigators.  The tables were intended to be viewed as separate landscape pages, but they were reformatted in portrait layout in the pdf proof.  Ideally, an overall view of the information will be easier to appreciate in the final version.

Comment 3.4: Beyond these recommendations concerning style, the article provides an accurate overview of the field and suggests new therapeutic approaches that are worthwhile and should be considered and published.

Response 3.4: Once again, thank you for the favorable feedback.

Reviewer 3 Report

The aim of the review was to give a comprehensive overview of the landscape of completed and ongoing clinical trials concerning the use of PARP inhibitors in gliomas.

In summary, it can be stated that the authors well highlighted the different aspects of possible combination therapies as well as the basics of the mechanism of action, the relevance of the blood brain barrier, and the need to stratify the patients for relevant (epi)genomic changes of the tumor cells to understand possible limitations to the clinical use of PARP inhibitors in different settings. It well summarizes the possible clinical use of PARP inhibitors but also relevant side effects of distinct combination therapies and thus outlines new therapy regimes that could be promising for distinct subgroups of the heterogeneous glioma patient cohort in the future.

Line 205-207: Please add a reference to the statement, that PARP inhibitors abrogate the upregulation of the DNA repair in glioma stem cells. It is shown that glioma stem cells also, for example, upregulate non-homologous end-joining which should not be directly affected solely by PARP inhibition. So the conclusion that PARP inhibition alone can abrogate the radioresistance of glioma stem cells via inhibiting the DNA repair seems questionable without adequate experimental reference.

Line 208-209: Please clarify your statement regarding tumor hypoxia as a mechanism of resistance against radiation. From the references you mentioned it is not clearly evident that there is any effect on the hypoxia in the tumor. Also I did not find the statement of the vasodilatory effect in the mentioned papers. The only statement is that PARP inhibition decreases angiogenesis, which would primarily cause more hypoxia (and of course later a lack of nutrients, too). But this is also a suggested key feature of glioma stem cells to deal with these situations. Additionally, less perfusion would also limit the distribution of the PARP inhibitors within the tumor and that would lead to a less effective PARP inhibition and thus abolish a potential radiosensitizing effect. So please check again and show the experimental evidence in the references for your statement.

Author Response

Reviewer #3

Comment 4.1: The aim of the review was to give a comprehensive overview of the landscape of completed and ongoing clinical trials concerning the use of PARP inhibitors in gliomas.  In summary, it can be stated that the authors well highlighted the different aspects of possible combination therapies as well as the basics of the mechanism of action, the relevance of the blood brain barrier, and the need to stratify the patients for relevant (epi)genomic changes of the tumor cells to understand possible limitations to the clinical use of PARP inhibitors in different settings.  It well summarizes the possible clinical use of PARP inhibitors but also relevant side effects of distinct combination therapies and thus outlines new therapy regimes that could be promising for distinct subgroups of the heterogeneous glioma patient cohort in the future.

Response 4.1: Thank you for the favorable feedback.

Comment 4.2: Line 205-207: Please add a reference to the statement, that PARP inhibitors abrogate the upregulation of the DNA repair in glioma stem cells.  It is shown that glioma stem cells also, for example, upregulate non-homologous end-joining which should not be directly affected solely by PARP inhibition.  So the conclusion that PARP inhibition alone can abrogate the radioresistance of glioma stem cells via inhibiting the DNA repair seems questionable without adequate experimental reference.

Response 4.2: Thank you for the clarification.  Glioma stem cells can upregulate DNA damage repair systems as a resistance mechanism.  There is emerging evidence that PARP inhibitors may contribute to abrogating this response.  We have reworded our statement and included a supporting reference (please refer to PARP inhibition with radiotherapy or tumor treating fields: Paragraph 1).

Comment 4.3: Line 208-209: Please clarify your statement regarding tumor hypoxia as a mechanism of resistance against radiation.  From the references you mentioned it is not clearly evident that there is any effect on the hypoxia in the tumor.  Also I did not find the statement of the vasodilatory effect in the mentioned papers.  The only statement is that PARP inhibition decreases angiogenesis, which would primarily cause more hypoxia (and of course later a lack of nutrients, too).  But this is also a suggested key feature of glioma stem cells to deal with these situations.  Additionally, less perfusion would also limit the distribution of the PARP inhibitors within the tumor and that would lead to a less effective PARP inhibition and thus abolish a potential radiosensitizing effect.  So please check again and show the experimental evidence in the references for your statement.

Response 4.3: Thank you for the clarification.  PARP inhibitors exerted a vasodilatory effect in preclinical tumor models, which reduced tumor hypoxia and radioresistance.  They may also have separate anti-angiogenic effects which enhance radiation sensitivity.  We have reworded our statement and included a supporting reference.

Reviewer 4 Report

Very well-written paper. The manuscript is comprehensive and can be accepted in its current form. 

Author Response

Reviewer #4

Comment 5.1: Very well-written paper. The manuscript is comprehensive and can be accepted in its current form.

Response 5.1: Thank you for the favorable feedback.

Round 2

Reviewer 1 Report

Much improved. Accept for publication in its present form